# The Optimization of Secondary Lining Construction Time for Shield Tunnels Based on Longitudinal Mechanical Properties

Shaobo Chai [1], Yifan Yan [1], Bo Hu [1,2,*], Hongchao Wang [3], Jun Hu [1,2], Jian Chen [2,4], Xiaodong Fu [2,4] and Yongqiang Zhou [2,4]

[1] School of Civil Engineering, Chang'an University, Xi'an 710064, China; shbchai@chd.edu.cn (S.C.)
[2] State Key Laboratory of Geomechanics and Geotechnical Engineering, Institute of Rock and Soil Mechanics, Chinese Academy of Sciences, Wuhan 430071, China
[3] China Railway Eryuan Engineering Group Co., Ltd., Chengdu 610031, China
[4] School of Engineering Science, University of Chinese Academy of Sciences, Beijing 100049, China
* Correspondence: hubochd@163.com

**Abstract:** In the field of shield tunnels, the occurrence of uneven longitudinal settlement in segment linings has presented persistent challenges, including heightened risks of localized damage and water leakage. While the adoption of a secondary lining has been proposed as a viable solution to these issues, the question of how to select an appropriate construction time for the secondary lining, one that enables it to fully harness its load-bearing capacity while optimizing the tunnel's overall stress and deformation characteristics, continues to be a pressing concern. To address this issue, this study established a three-dimensional longitudinal refined numerical model of double-layer-lined shield tunnel. In addition, the deformation degree of the segment lining was used as a time indicator to define the construction time for the secondary lining. Subsequently, an analysis of the impact of the construction time of the secondary lining on the longitudinal mechanical properties of the double-layer-lined shield tunnel is conducted through an assessment of tunnel longitudinal deformation and structural stress. The research findings indicated that the construction of the secondary lining improved the longitudinal deformation resistance of shield tunnels. Simultaneously, it led to a significant increase in the longitudinal shear forces within the segment lining and a notable reduction in longitudinal bending moments. Moreover, the construction time of the secondary lining played a pivotal role in these alterations. Considering the longitudinal force situations and load-bearing characteristics of the double-layer lining structure, it was determined that the optimal construction time for the secondary lining fell within the range of 20% to 40% of the total construction duration. In this scenario, the deformation and internal forces within the segment lining remained within permissible limits. Additionally, both the segment lining and the secondary lining were able to fully utilize their load-bearing capacities, ensuring the economic and safety aspects of the tunnel.

**Keywords:** shield tunnel; double-layered lining; refined numerical model; secondary lining construction time; longitudinal mechanical properties





## 1. Introduction

With the rapid advancement of underground transportation, the utilization of shield tunnels has expanded considerably [1]. A shield tunnel comprises a tubular structure connected by bolts, and the existence of joints compromises the tunnel's overall rigidity [2]. When exposed to factors like seismic events [3,4], fluctuations in water levels [5,6], and the excavation of adjacent tunnels [7,8], substantial longitudinal deformations can manifest within the tunnel, giving rise to potential concerns such as water seepage, structural deterioration, and localized damage. As an illustration, over the span of four years from 2005 to 2009, the maximum settlement value for Nanjing Metro Line 1 has reached 122 mm, coupled with noticeable structural cracking [9]. Likewise, in the construction of Shanghai

Metro Line 4, the permafrost surrounding the shield tunnel was disturbed, causing the swift infiltration of soil and water into the tunnel, ultimately resulting in substantial damage to the lining (2004) [10]. Similar accidents were also observed in Foshan (2018) [11], Kaohsiung (2007) [10], Tianjin (2011) [12] and Nanjing (2007) [13], all with disastrous consequences. All the large-scale failures in these shield tunnels originated from small-scale local failures. In response to these phenomena, some shield tunnel projects [14,15] adopted double-layered lining structures, using a secondary lining as a safety reserve or reinforcement measure to ensure the safety of the shield tunnel structure. Compared with single-layered shield tunnels, double-layered structures are applied in smaller areas and remain concentrated in large-diameter river-crossing or sea-crossing tunnels [16] and water transfer tunnels [17]. In the early double-layered shield tunnel projects, the segment lining was generally used as the load-bearing structure, while the secondary lining was used as an auxiliary structure to improve the durability and water resistance of the shield tunnel. For example, the Trans-Tokyo Bay Highway Tunnel used a secondary lining with a thickness of 35 mm as a structural safety buffer in the Tokyo Bay area with soft submarine strata and significant water pressure [18]. However, existing studies show [19] that the secondary lining can be used as one of the load-bearing structures in a shield tunnel to enhance the overall stiffness. Furthermore, with the development of urban rail transit, the shield tunnel is facing more complex engineering situations, and the design and usage requirements are constantly being improved. Compared with single-layered segment lining, a double-layered lining structure has more advantages in reinforcement, fire prevention, collision resistance, anti-settlement, anti-erosion, and water pressure resistance. Consequently, an increasing number of shield tunneling projects, such as the Rail-cum-Road Yangtze River Cross Tunnel in Wuhan [20], the Shiziyang Tunnel [21], and the Wusongkou Yangtze River-crossing tunnel in Shanghai [22], are opting for double-layered lining structures. Therefore, it is important to study the effect of the secondary lining on the mechanical properties of such shield tunnels.

Previous studies have analyzed the mechanical properties of double-layered shield tunnels mainly through model experiments, theoretical analysis, and numerical simulation methods. Among them, the model experiment is an intuitive and reliable method for studying the mechanical properties of structure. Feng et al. [23] investigated the differences in the mechanical properties of underwater shield tunnels between single-layered lining and double-layered lining structures through models and field tests. In terms of the theoretical method, Ei et al. [24] proposed an analytical analysis method for double-layered linings that can consider both joint effects and the interactions between soil and lining, as well as lining and lining. Wu et al. [25] presented a new longitudinal structural model considering the shear-dislocation between rings, and proposed an equivalent shear stiffness to consider the influence of joints. As a more comprehensive, cost-effective, and efficient calculation method, numerical simulation has become increasingly important in studying the mechanical properties of double-layered shield tunnels. Based on double-layered lining joint load tests, Zhang et al. [26,27] pioneered three contact surface connection models of double-layered lining according to different simplifications of the contact surface of the double-layered lining. Yan et al. [28,29] proposed a contact surface combined model of a double-layered lining with a compression bar and shear spring on the basis of the shear compression resistance model after comparing and analyzing the differences in mechanical properties of the above three models. Most of the subsequent studies of double-layered lining are based on the combined model with a compression bar and shear spring. For example, Liang et al. [30] established a three-dimensional double-layered lining model of shield tunnels with segments and joints, and simulated the contact surface with a shell, spring, and a combination of a compression bar and shear spring. Wang et al. [31] constructed a calculation model of double-layered shield tunnels that can reflect the non-linearity of the flexural stiffness of the joint and the failure mechanism of compression-shear spring of contact surface between segment and secondary lining. However, the above

studies focused on the analysis of lateral mechanical properties of double-layered shield tunnels and ignored the influence of the longitudinal mechanical properties.

Previous studies [32] found that the presence of segments staggered assembly and bolt joints would reduce the overall longitudinal stiffness of shield tunnels and renders the structure more flexible. When the geological conditions change, the surrounding soil is excavated, or the ground is stacked unevenly [33,34], the shield tunnels are susceptible to uneven longitudinal settlements, local damage being induced to segments, and water leakage in joints. Therefore, the longitudinal mechanical properties of double-layered shield tunnels deserve attention. In recent years, some scholars have studied the longitudinal equivalent bending stiffness and deformation distribution of double-layered shield tunnels using theoretical methods or numerical simulation, but the studies have mostly focused on the effects of load, variation of material property and thickness of the secondary lining [35–37]. The application of the secondary lining at a reasonable time can better improve the overall internal force distribution and give full play to the structural load-bearing capacity of doubled-layered shield tunnels [38]. However, previous studies [39,40] on the construction time of the secondary lining mainly take a single segment ring or several representative segment rings as the research object to investigate the lateral mechanical properties of segment lining structure; however, they did not consider the longitudinal mechanical properties, and ignore the adverse effects of the longitudinal deformation of shield tunnels on their overall performance, thus possessing certain limitations.

This study aims to investigate the influence of the construction time of the secondary lining on the longitudinal mechanical properties of the double-layered lining of shield tunnels, as well as to determine the optimal construction time for the secondary lining. To achieve these objectives, numerical simulation methods are employed to analyze the distribution of internal forces and deformations within shield tunnels under different construction time scenarios for the secondary lining. By considering the load-bearing characteristics of various sections in the double-layered shield tunnel, the study identifies the most suitable construction time for the secondary lining. The research methodology is illustrated in Figure 1.

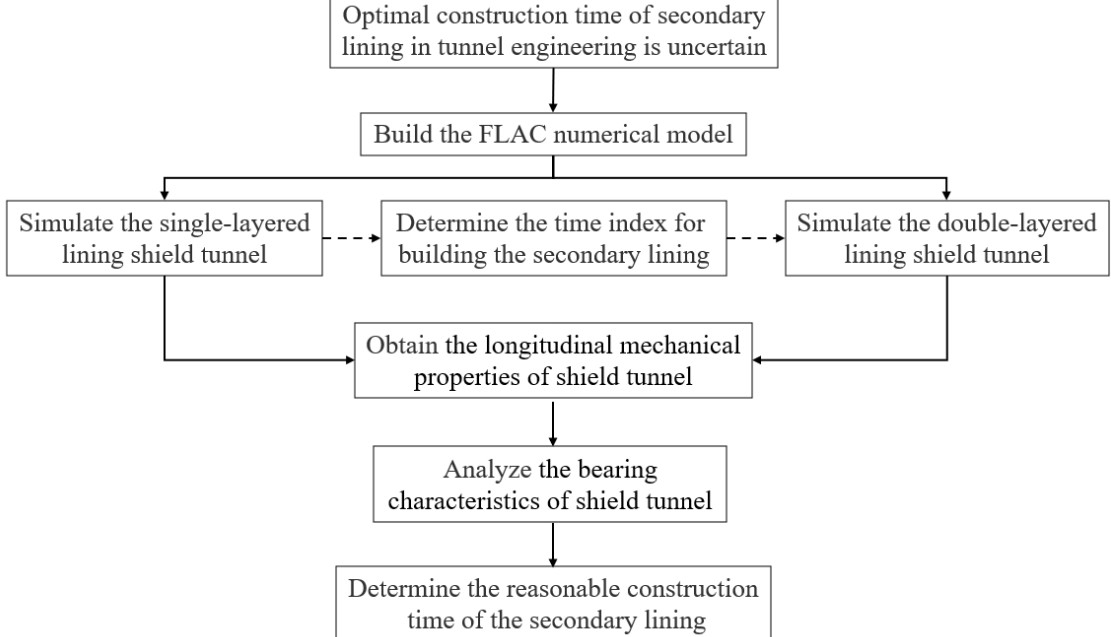

**Figure 1.** The research methodology for this study.

## 2. Numerical Approach

### 2.1. Engineering Context

The Nantong Haitai River-Crossing Project in Jiangsu Province, China was chosen for analysis; its total length was 11,360 m, including a shield section of 9250 m. The cross-section of the shield tunnel is illustrated in Figure 2. It possessed an outer diameter of 14.5 m and an inner diameter of 12.7 m, categorizing it as an ultra-large-diameter underwater shield tunnel. The thicknesses of the segment and the secondary lining were 600 mm and 300 mm, respectively. Figure 3 provides an overview of the geological conditions along the longitudinal axis of the tunnel. As delineated in the figure, the tunnel traversed the expanse of the Yangtze River, with a tunnel segment extending for a distance of 7200 m beneath the water surface. The maximum depth of the river reached 24.5 m. At the midpoint of the river crossing, the tunnel structure encountered the highest hydraulic head, measuring at 71.63 m, while the maximum hydraulic pressure was documented at 0.72 MPa. These conditions imposed substantial demands on the tunnel's waterproofing measures. Moreover, the tunnel was located within an intricate geological setting. It traversed a range of soil layers, encompassing silty clay, silty fine sand, and muddy silty clay, with significant variations in the mechanical properties of these soil layers. Along the tunnel's trajectory, there were marked fluctuations in both overburden thickness and water pressure, initially increasing and subsequently decreasing. These factors had the potential to induce non-uniform longitudinal deformations within the tunnel, thereby jeopardizing its safety. Consequently, this river-crossing shield tunnel project holds considerable significance as a point of reference for the investigation of the longitudinal mechanical characteristics of a double-layered shield tunnel.

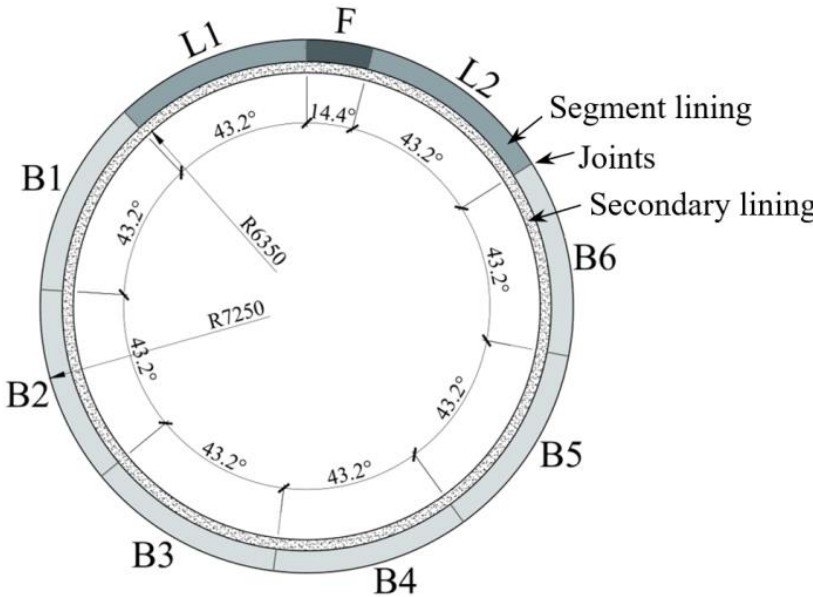

**Figure 2.** Composition and dimensions of double-layered shield tunnel.

### 2.2. The Numerical Model

This paper utilizes FLAC$^{3D}$, which is based on the finite difference method, to perform calculations. Geotechnical engineering problems are typically expressed as differential equations, including basic equations such as the balance equation, geometric equation, and constitutive equation, as well as boundary conditions. By replacing each derivative with a finite difference approximation formula, the finite difference method transforms the problem of solving partial differential equations into the problem of solving algebraic equations. FLAC$^{3D}$ takes nodes as the calculation objects and concentrates both forces and

masses at each node, then solves them in the time domain using the equation of motion. The equation of node motion can be expressed as follows:

$$\frac{\partial v_i^l}{\partial t} = \frac{F_i^l(t)}{m^l} \tag{1}$$

where $F_i^l(t)$ is the unbalance force component of node $l$ at time $t$ in direction $i$, which can be derived from the principle of virtual work. $m^l$ is the concentrated mass of node $i$. FLAC$^{3D}$ uses the velocity to calculate the strain increment of the element in a time step, as follows:

$$\Delta e_{ij} = \frac{1}{2}(v_{i,j} + v_{j,i})\Delta t \tag{2}$$

where $v_{i,j}$ is the partial derivative of $v$ with respect to the vector component $x_i$ of the node position. By using the constitutive equation, the stress increment can be obtained from the strain increment. Then, by accumulating the stress increment of each time step, the total stress can be obtained.

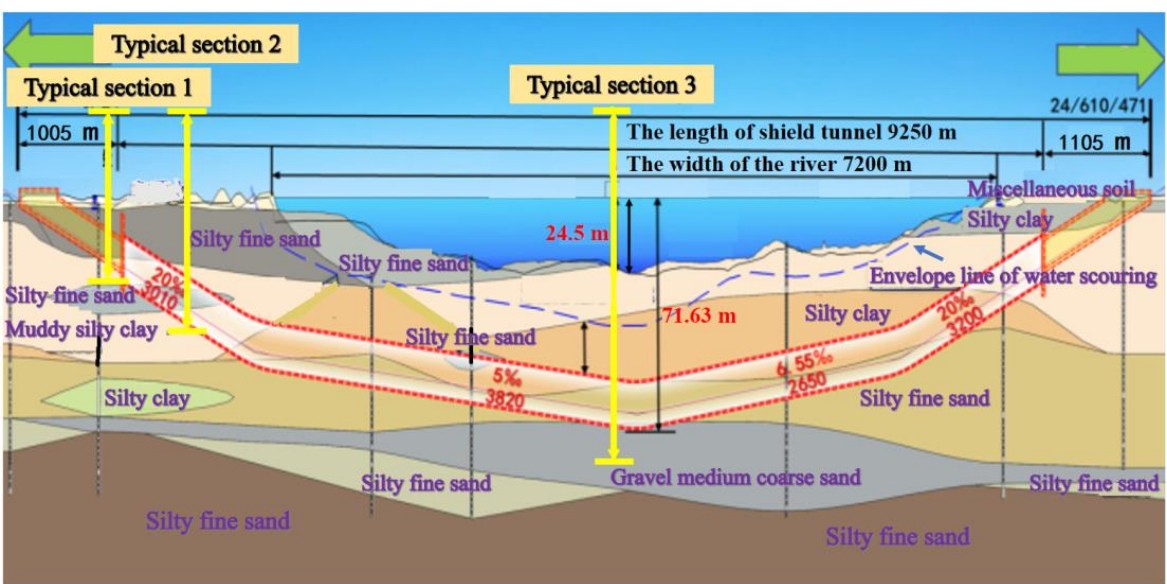

**Figure 3.** Schematic diagram of geological cross-section along the tunnel's longitudinal axis.

The aforementioned shield tunnel project was utilized as an exemplar for the formulation of a numerical model, depicted in Figure 4. The model's dimensions were 36.1 m × 100 m × 100 m (X × Y × Z). The lateral side of the model represented the boundary with a normal displacement constraint, while the bottom surface served as the fully constrained boundary. As this paper primarily focused on the mechanical properties of shield tunnel structures rather than the effect of strata deformation or strata properties, the Mohr–Coulomb constitutive model, known for its simplicity and wide applicability [41,42], was employed for the strata. Table 1 presents the specific physical-mechanical parameters for the strata. The shield tunnel was located at the center of the model. Tunnel excavation was simplified to the excavation of the entire section at once. In the shield tunnel project upon which this paper relies, the soil cover near the riverbed was thicker, and the water head was higher. Consequently, the tunnel at the riverbed experienced a higher water and soil load compared to the adjacent sides. To streamline modeling and expedite model calculations, a trapezoidal stress boundary was implemented at the upper part of the model to represent the actual loading conditions. Within the model, $q_1$ and $q_2$ represented the soil and water load at the typical sections 1 and 3, respectively.

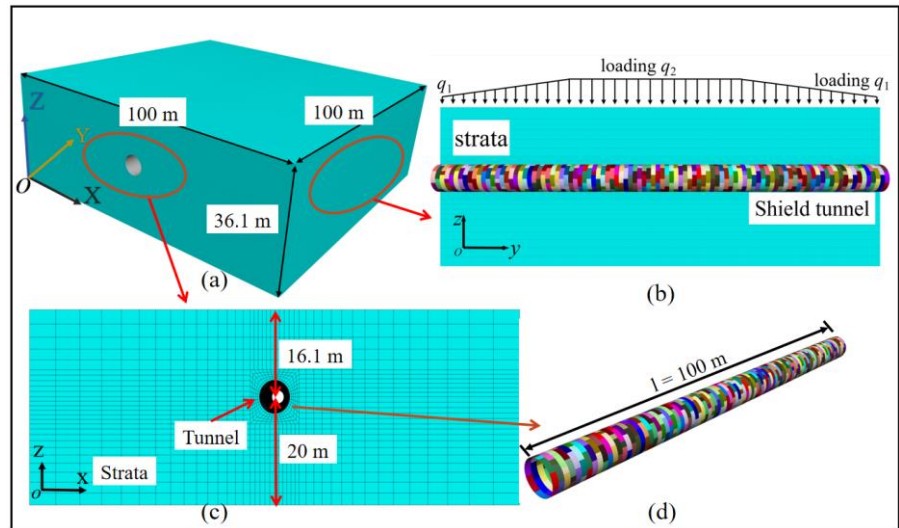

**Figure 4.** (**a**–**d**) Dimensions and boundary conditions of the tunnel model.

**Table 1.** Physical and mechanical parameters of stratum.

| Volumetric Weight (kN/m³) | Internal Friction Angle (°) | Cohesive Force (kPa) | Elastic Modulus (MPa) | Poisson Ratio |
|---|---|---|---|---|
| 18.1 | 30 | 10 | 26.18 | 0.3 |

Combined with the real engineering project, a three-dimensional longitudinal refined numerical calculation model of a double-layered shield tunnel was established in the present research, and the longitudinal mechanical properties of the shield tunnel were studied. The established model was a double-layered shield tunnel with 100 rings and a total length of 100 m. A single segment ring consists of six standard blocks, two adjacent blocks, and one capping block, and the segments were assembled in a staggered manner. The secondary lining was placed inside the segment lining and composed of several homogeneous secondary lining rings (Figure 5).

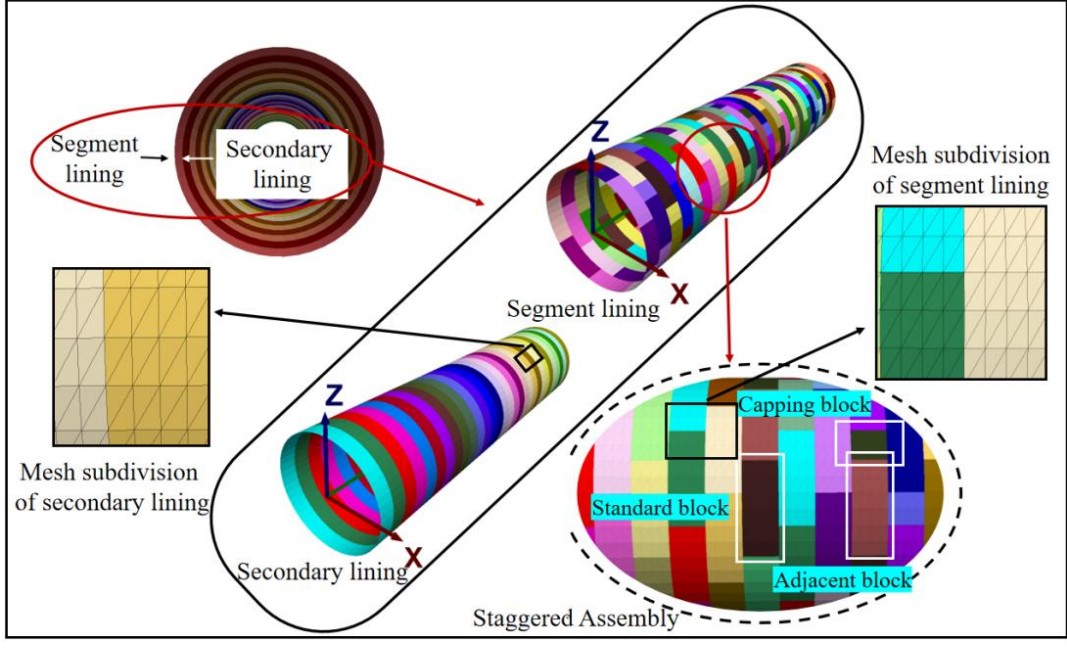

**Figure 5.** The assembly method and mesh division of the double-layered shield tunnel model.

The refined numerical calculation model was calculated based on the finite difference method, and the segment lining was simulated using the Liner element (a three-node flat finite element that can resist the shear force, axial force, and bending moment, as well as simulate the effects of tangential friction and the tension and compression in the normal direction on the contact surface between the segment lining and the secondary lining). The thickness, stiffness, and strength of the secondary lining are relatively small, and the mechanical mechanism is in accordance with the characteristics of thin shell structure; therefore, the shell element was utilized to simulate the secondary lining. The parameters of segment lining and secondary lining were determined based on the engineering data shown in Figure 2, as listed in Table 2.

**Table 2.** Parameters of lining.

| Lining Type | Concrete Strength Grade | Thickness/mm | Elastic Modulus/GPa | Poisson's Ratio |
|---|---|---|---|---|
| Segment lining | C50 | 60 | 34.5 | 0.25 |
| Secondary lining | C30 | 30 | 30 | 0.25 |

For the liner element used to simulate the segment, a reasonable method was used to divide the mesh so that the structural nodes were located exactly where the longitudinal and circumferential bolts are. This treatment allowed for the better implementation of the refined simulation of the bolts later. In addition, for shell elements used to simulate the secondary lining, the nodes of the mesh needed to coincide with the liner element. In this way, a reasonable interface element could be established between the liner element and shell element, so as to realize a reasonable simulation of the contact surface between the segment lining and the secondary lining. For convenience, the same meshing as used for the liner element was used for the shell element. The specific meshing of the model is shown in Figures 4 and 5. The entire model comprised a total of 536,400 solid elements, 556,187 grid points, 70,400 structural elements, and 43,644 structural nodes.

The segments were assembled in a staggered manner in the refined numerical calculation model. They were joined into a segment ring using circumferential bolts, then connected into a longitudinal 3D tunnel structure using longitudinal bolts. Link elements were employed to simulate the bolts. Making the connection position and connection characteristics of the link element align with bolts used in practice helped to reflect the deformation mechanisms of the shield tunnel realistically. Each segment was uniformly divided into four pieces along the width direction, and two adjacent segments in same segment ring had five pairs of nodes with overlapping positions at the connection location of segments, numbered 1 to 5. The link element was established at nodes 2 and 4 to simulate the circumferential bolts (Figure 6a). Then, each segment ring was uniformly divided into 50 pieces along the circumferential direction and there were 50 nodes with an overlapping position between two adjacent segment rings. The link element was established at every other node to simulate the longitudinal bolts, with a total of 25 link elements, as shown by the red dots in Figure 6b. There were three link elements on each standard and adjacent block segment and only one link element on each capping block segment.

Each link element has six degrees of freedom in six directions, including three translational degrees of freedom (one in each axial direction) and three rotational degrees of freedom (one around each axis). Each degree of freedom has independent one-dimensional constitutive relationships and mechanical properties. Previous studies showed that the flexural stiffness $K_R$, axial stiffness $K_C$, and shear stiffness $K_T$ of the link element exert a more significant influence on the mechanical properties of the shield tunnel. So, the effects of the three degrees of freedom related to the aforementioned stiffness parameters were mainly considered in the present research, which were set as elastic connections and represented by springs. The remaining three degrees of freedom were set as rigid connections. The specific connection form is shown in Figure 7a, which is a schematic representation of the

composition of three elastic springs for any link element. The linear yield ideal elastoplastic constitutive relation illustrated in Figure 7b was adopted for springs on the three degrees of freedom [43], and it can be expressed as:

$$\sigma_b = \begin{cases} K \cdot \varepsilon_b & (0 \le \varepsilon_b < \varepsilon_{b0}) \\ \sigma_{bf} & (\varepsilon_{b0} \le \varepsilon_b < \varepsilon_{bu}) \end{cases} \tag{3}$$

where $\sigma_b$ and $\varepsilon_b$ represent the stress and the strain of the spring, respectively; $\sigma_{bf}$ is the yield stress of spring; $\varepsilon_{b0}$ is the strain corresponding to $\sigma_{bf}$; $\varepsilon_{bu}$ is the ultimate strain of spring; $K$ denotes the stiffness of any spring from the three degrees of freedom with elastic connection.

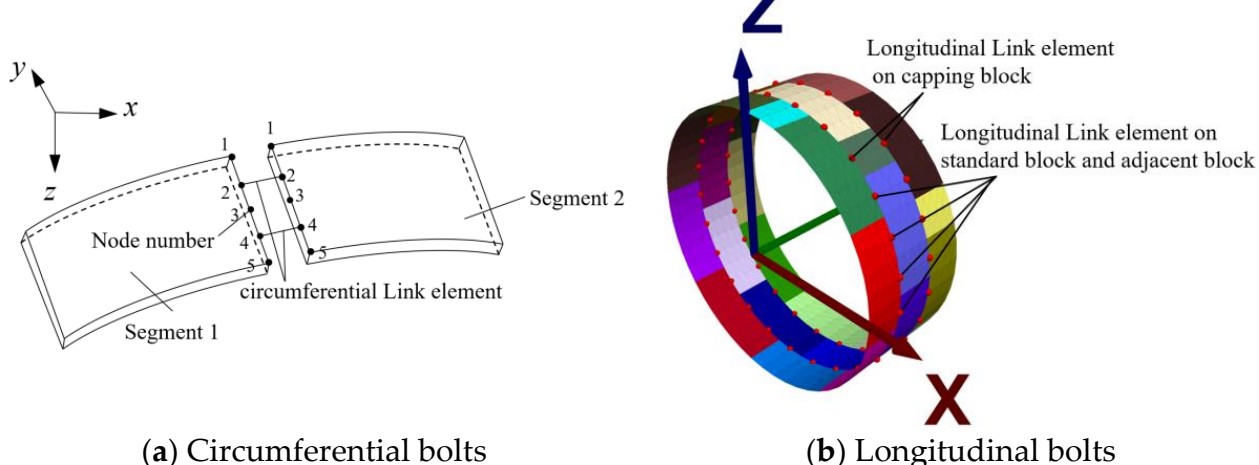

(**a**) Circumferential bolts  (**b**) Longitudinal bolts

**Figure 6.** Simulation method of bolts.

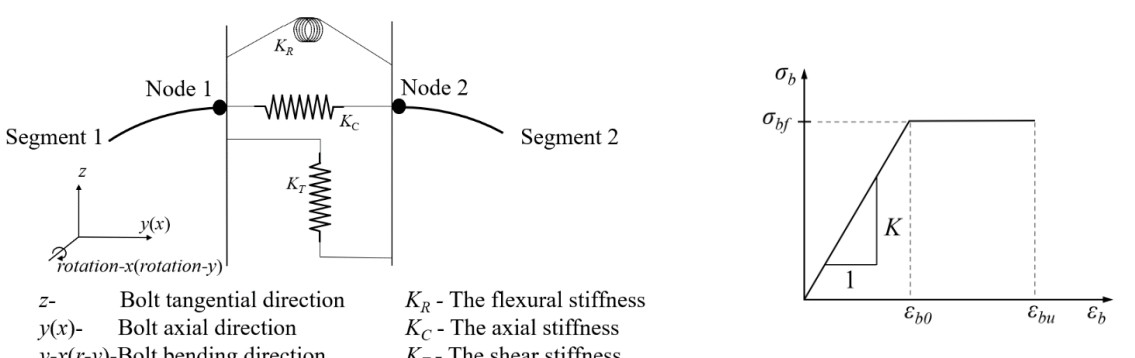

| z- | Bolt tangential direction | $K_R$ - The flexural stiffness |
| y(x)- | Bolt axial direction | $K_C$ - The axial stiffness |
| y-x(r-y)- | Bolt bending direction | $K_T$ - The shear stiffness |

(**a**) Characteristics of the link element  (**b**) Constitutive relationship of the spring

**Figure 7.** Bolt characteristics.

The spring stiffness was determined by the empirical values from the study [44], and the yield strength was a macroscopic strength parameter in the actual project, which was determined in conjunction with the findings of the present research and previous experimental studies [21], as shown in Table 3.

**Table 3.** Parameters of bolts between segments.

| Rotational Stiffness $K_R$/(MN·m·rad$^{-1}$) | Axial Stiffness $K_C$/(MN·m$^{-1}$) | Shear Stiffness $K_T$/(MN·m$^{-1}$) | Yield Strength of Spring $\sigma_{bf}$/(MPa) |
|---|---|---|---|
| 100 | 1050 | 500 | 640 |

The contact characteristics of the contact surface between the segment lining and the secondary lining and their parameter value exert an important influence on the participation extent of the secondary lining in the structural bearing process. Previous studies have shown [23] that the contact characteristics should include contact and separation in the normal direction as well as adhesion and friction in the tangential direction. Except for the tension and pressure in the normal direction, the contact surface can also transmit a certain shear force in the tangential direction. The contact characteristics in the refined numerical calculation model consist of a linear spring with tensile strength in the normal direction and two linear yield springs in the tangential direction, which were implemented using the interface element, as shown in Figure 8a. Figure 8b shows the shear strength criterion for the interface element, where the compressive strength of the interface element too high so its compressive failure was ignored. The maximum shear stress $\tau_{max}$ changes with the normal stress $\sigma_n$, as shown in Equation (4):

$$\tau_{max} = \begin{cases} c_r & (\sigma_n < -f_t) \\ c & (-f_t \leq \sigma_n < 0) \\ c + \sigma_n \tan \varphi & (0 \leq \sigma_n) \end{cases} \tag{4}$$

where $\tau_{max}$ is the maximum stress of the shear spring of the interface element; $\sigma_n$ refers to the stress of normal spring of the interface element; $c$ and $c_r$ represent the cohesion and the residual cohesion of the interface element in the tangential direction, respectively; $\varphi$ is the friction angle of the interface element in the tangential direction; and $f_t$ is the tensile strength of the normal spring of the interface element.

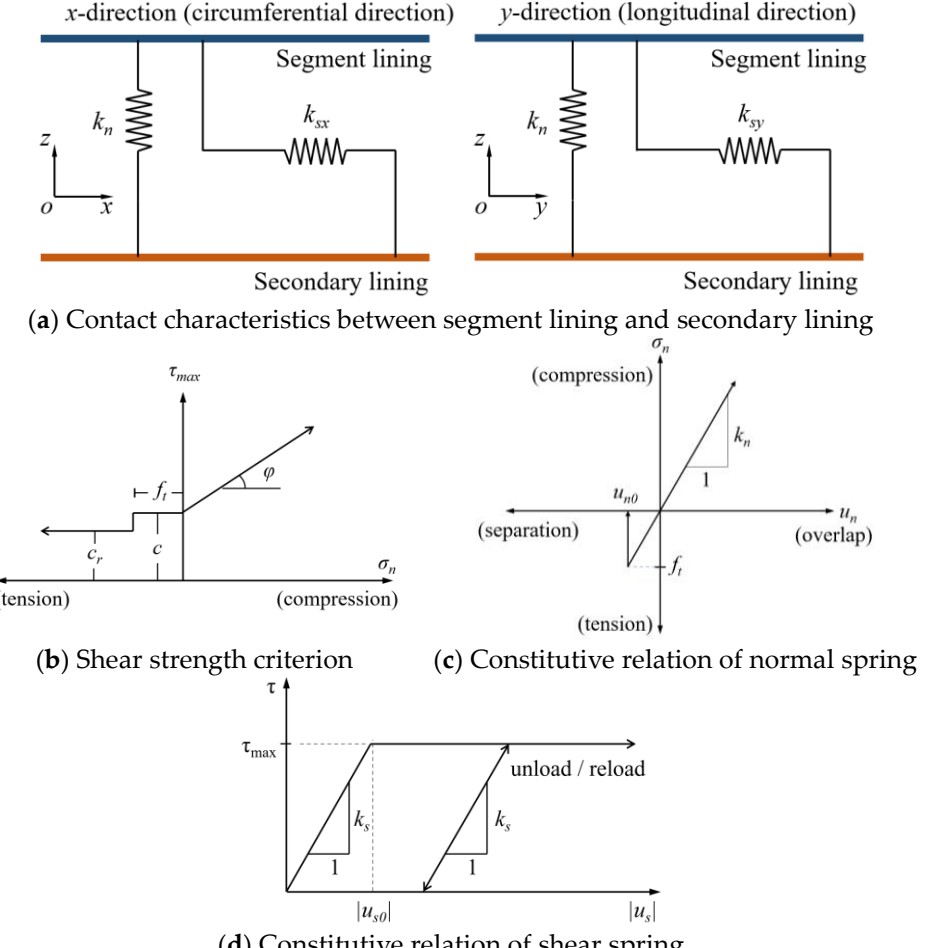

(**a**) Contact characteristics between segment lining and secondary lining

(**b**) Shear strength criterion     (**c**) Constitutive relation of normal spring

(**d**) Constitutive relation of shear spring

**Figure 8.** Characteristics of the interface element.

The constitutive relationship governing the behavior of the normal and shear spring is shown in Figure 8c and d, respectively. When the normal spring is compressed, the maximum stress of shear spring $\tau_{max}$ increases continuously with the stress on the normal spring $\sigma_n$. When the behavior of the normal spring is tensile, the maximum stress of the shear spring $\tau_{max}$ is equal to the cohesion $c$. If the stress reaches the tensile strength $f_t$, the normal spring of the interface element will undergo tensile failure. The cohesion $c$ is then replaced by the residual cohesion $c_r$, and the tensile strength $f_t$ is reduced to zero. The behavior of the shear spring of the interface element will return to its original state if the segment lining and secondary lining overlap and the normal spring generates pressure again. Based on these deformation properties, this refined numerical calculation model can not only reflect the shear behavior, but also realistically simulate the separation and subsequent overlap between segment lining and secondary lining. The constitutive relationship governing the behavior of the normal and shear spring can be expressed thus:

$$\sigma_n = k_n u_n \quad (u_{n0} \leq u_n) \tag{5}$$

$$\tau = \begin{cases} k_s|u_s| & (0 \leq u_s < |u_{s0}|) \\ \tau_{max} & (|u_{s0}| < u_s) \end{cases} \tag{6}$$

where $\tau$ is the stress of the shear spring of the interface element; $|u_s|$ is the absolute strain in the shear spring of the interface element; $|u_{s0}|$ is the strain in the shear spring corresponding to the maximum shear stress $\tau_{max}$; $u_n$ denotes the strain in the normal spring of the interface element; $u_{n0}$ is the strain in the normal spring corresponding to the tensile strength $f_t$; and $k_s$ and $k_n$ represent the stiffness of the shear and normal spring of the interface element, respectively.

The stiffness of the interface element in a circular tunnel can be calculated empirically using Equation (7), as suggested in the literature [43,45]:

$$k_n = k_s = 100 \max \left[ \frac{\left( K + \frac{4}{3}G \right)}{\Delta z_{min}} \right] \tag{7}$$

where $K_z$ and $G_z$ represent the bulk and shear modulus of adjoining zone, respectively; $\Delta z_{min}$ is the smallest dimension of an adjoining zone in the normal direction. Parameters such as tensile strength $f_t$, cohesion $c$ and $c_r$, and friction $\varphi$ were chosen from the empirical values given in the literature, as summarized in Table 4.

**Table 4.** Parameters of the interface between segment lining and secondary lining.

| Normal Stiffness $k_n/(\mathrm{MN \cdot m^{-1}})$ | Tangential Stiffness $k_s/(\mathrm{MN \cdot m^{-1}})$ | Normal Tensile Strength $f_t/\mathrm{MPa}$ | Cohesion $c/\mathrm{MPa}$ | Residual Cohesion $c_r/\mathrm{MPa}$ | Friction Angle $\varphi/°$ |
|---|---|---|---|---|---|
| 34,900 | 14,500 | 4 | 4 | 4 | 20 |

## 3. Analysis of the Construction Time of the Secondary Lining

The construction time of the secondary lining exerts a significant influence on the safety and efficacy of a double-layered shield tunnel. In previous experimental studies, the inverse method was generally used to obtain the external loads corresponding to different times of construction, and then the external load was applied to make the segment lining reach a predetermined deformation, after which the secondary lining was activated and the full load was applied. However, in the actual shield tunnel project, the load on the structure does not increase gradually, and this method was difficult to implement. It is worth noting that the deformation of tunnel increases gradually with time. Therefore, the deformation of segmental lining was used to assess the optimal construction time of the secondary lining (using the ratio between the vault deformation of segment lining ring at the moment of the

secondary lining constructed and the final deformation of the vault of the single-layered shield tunnel, taking the middle segment ring of the tunnel as a reference). The formula can be expressed as:

$$k = \frac{\eta'}{\eta} \tag{8}$$

Firstly, the final vault deformation $\eta$ of a single-layered shield tunnel in the middle segment ring was calculated, and then the predetermined deformation $\eta'$ under different construction times of the secondary lining was inverted using Equation (8). During the calculation, the vault deformation of the middle segment ring was monitored and the secondary lining was activated when the deformation reaches the predetermined deformation $\eta'$, and then the calculation continues until convergence. In shield tunnel construction, the segments are installed shortly after tunnel excavation and contribute to the overall force of the tunnel. The secondary lining is typically constructed with early strength cement concrete, which quickly gains strength after pouring and provides load-bearing capacity. Therefore, it is reasonable for the calculation model to simplify the construction process of segment and secondary lining. The effects of different construction times of the secondary lining on the longitudinal mechanical properties of the double-layered shield tunnel were analyzed on that basis, and the reasonable construction time was determined. In practice, the vault deformation of the segment ring can be monitored, while the final deformation of the vault of a single-layered shield tunnel can be predicted using the refined numerical simulation method.

### 3.1. Criteria for Determining the Construction Time of the Secondary Lining

In practical engineering, the deformation forms of the segment lining vault included vertical deformation (settlement) and dislocation. Therefore, it was necessary to select an appropriate deformation type to determine the construction time of the secondary lining. The deformation form needed to meet the following characteristics:

(1).   The deformation range was large enough to ensure sufficient resolution corresponding to different secondary lining construction times.
(2).   Deformation needed to be easy to measure, meaning it was necessary to consider the construction characteristics of the actual shield tunnel and the challenges associated with deformation measurement.

In practical engineering, the vertical deformation of the vault of a single-layered shield tunnel can be obtained by conventional monitoring means, while the final vertical deformation can be predicted using the refined numerical simulation method. The vertical deformation cloud map of the single-layered shield tunnel in the stratum was displayed in Figure 9. The segment rings were numbered from 1 to 100 (left to right), and the vault vertical deformation at the midpoint of the width of each ring was used to create the tunnel longitudinal deformation curve, as shown in Figure 10. Due to the symmetry of the model, the vertical deformation of the shield tunnel was also symmetrical. When subjected to a non-uniform load, the vertical deformation of the shield tunnel became notably non-uniform. Varied degrees of vertical deformation occurred in distinct sections of the tunnel. At both ends of the shield tunnel, the vertical deformation of the tunnel was maintained at a low level, measuring only about 2.5 mm. This was due to the constraints imposed by the boundary conditions and the minimal soil and water load resulting from the shallow burial depth of the tunnel. Subsequently, the soil and water load uniformly increased along the tunnel's direction, leading to a significant surge in the vertical deformation of the tunnel at this stage. In the central segment of the tunnel, the soil and water load reached its maximum and remained constant, resulting in a gradual rise in the vertical deformation of the tunnel along its longitudinal axis. The highest value was attained at the 40th ring of the shield tunnel, stabilizing at around 30 mm.

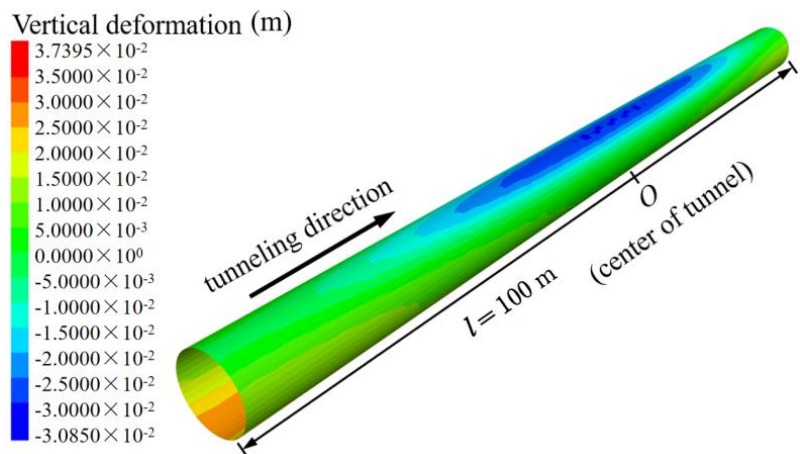

**Figure 9.** Vertical deformation cloud of the single-layered shield tunnel.

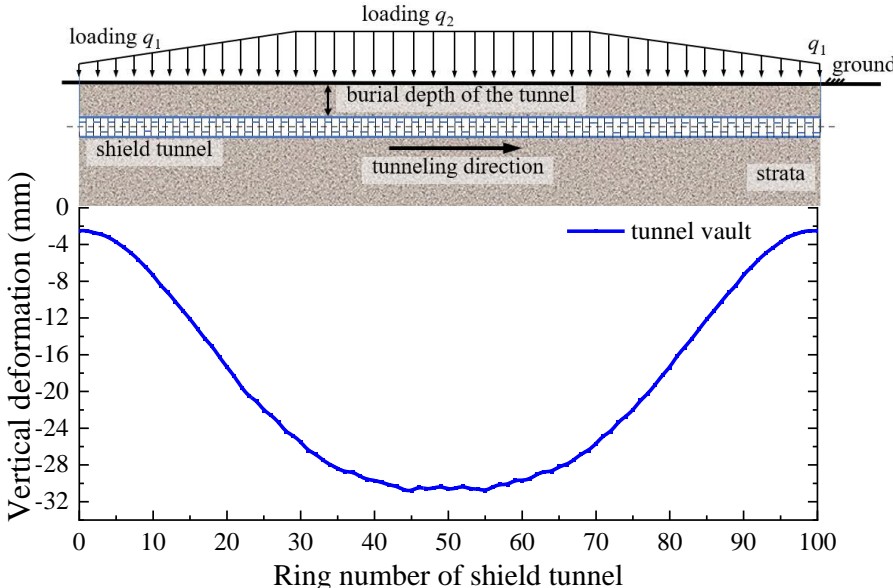

**Figure 10.** Longitudinal deformation curve of the single-layered shield tunnel.

When the shield tunnel was subjected to uneven loads, the segment lining experienced uneven settlement and the joints of adjacent segments would become misaligned. This kind of differential displacement between segments was referred to as dislocation. The dislocation of segment rings was often adopted to reflect the local deformation, and its variation along the longitudinal direction was also an important indicator of the longitudinal deformation of a shield tunnel. As shown in Figure 11, the dislocation of segment rings was the difference between the vertical deformation of adjacent segment rings, which could be considered the first order derivative of the vertical deformation distribution curve of the tunnel. It could also be interpreted as the rate of change of the vertical deformation, which was often used to reflect the degree of uneven deformation of the shield tunnel.

Figure 12 shows the longitudinal distribution of segment dislocation. It was observed that the dislocations of the segments exhibited clear discretization along the longitudinal direction. The distribution curve, as a whole, presented a cubic function distribution, and the specific fitting formula is: $y = 0.0000227x^3 - 0.00341x^2 + 0.110x + 0.198$. Within the 0–30 and 70–100 rings of the segment lining, the segment dislocation reached a maximum value of 1.81 mm. This was due to the gradual increase in load on both sides of the tunnel, leading to the pronounced uneven deformation of the segment lining and resulting in significant dislocation deformation. In the middle of the tunnel, i.e., within the 30–70 ring segments, the more uniform load on the segment rings led to the overall co-deformation of

the segment lining. During this stage, the segment's dislocation mostly remained under 0.5 mm, resulting in a relatively slight dislocation deformation. Additionally, in the regions marked as a, b, and c on the figure, the variation trend of segment dislocation was entirely consistent, and the distribution curve exhibited clear periodicity. This consistency was due to the staggered assembly of segments and a certain angular difference between adjacent segment rings. Specifically, the 11 segment rings formed a complete cycle.

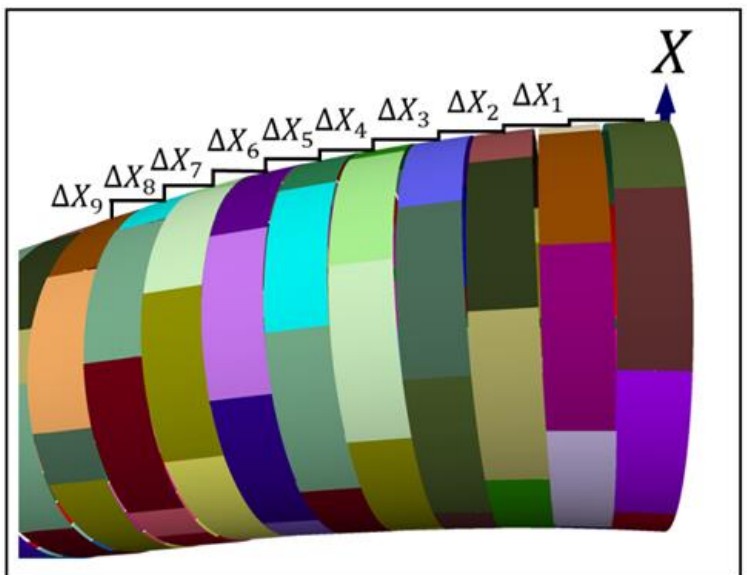

**Figure 11.** Schematic diagram of dislocation of segment rings.

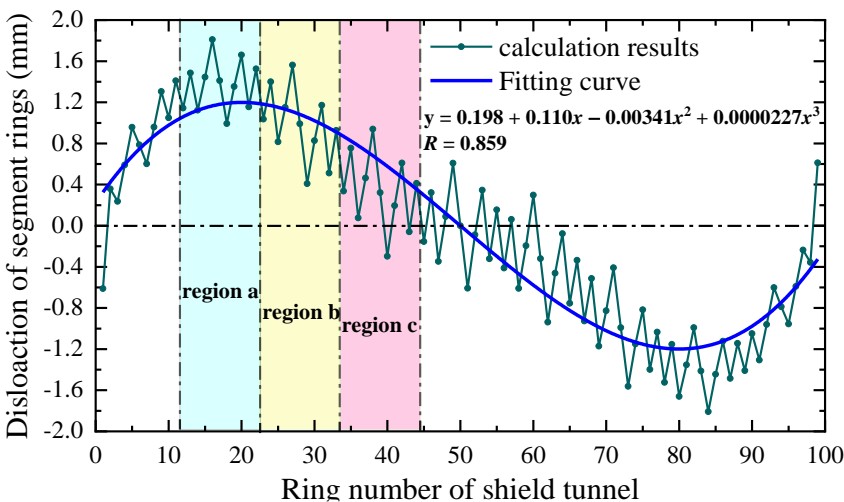

**Figure 12.** Longitudinal distribution curve of dislocation of segment rings.

## 3.2. Determination of the Construction Time of the Secondary Lining

Upon comparing the two deformation forms mentioned above, it became evident that the vertical deformation had a significantly larger range compared to dislocation deformation. This implied that when deciding on the construction time of the secondary lining, utilizing vertical deformation as the assessment criterion offered greater precision. Furthermore, in practical engineering, monitoring vertical deformation proved to be noticeably less challenging than monitoring dislocation deformation. Additionally, dislocation deformation was influenced by factors like segment manufacturing errors and assembly inaccuracies. Hence, the vertical deformation of the middle segment ring in the single-layered tunnel served as the foundation for determining the construction time of the secondary

lining. Levels of 0%, 10%, 20%, 30%, 40%, 50%, 60%, 70%, 80%, 90%, and 100% of the final vertical deformation were adopted as the 11 construction times of the secondary lining, and the predetermined deformations $\eta'$ corresponding to different construction times were calculated as shown in Table 5. Among them, the 0% and 100% construction time represents the secondary lining and segment lining were applied simultaneously and the secondary lining was activated after the tunnel deformation was stabilized, respectively.

**Table 5.** Vertical deformation values at different construction times (mm).

| Final Vault Vertical Deformation of the Single-Layered Shield Tunnel $\eta$/mm | The Vault Vertical Deformation of Segment Lining Ring upon Construction of the Secondary Lining $\eta'$/mm | | | | | | | | | | |
|---|---|---|---|---|---|---|---|---|---|---|---|
| | Construction Time of the Secondary Lining | | | | | | | | | | |
| | 0.0% | 10% | 20% | 30% | 40% | 50% | 60% | 70% | 80% | 90% | 100% |
| 30.85 | 0.00 | 3.09 | 6.17 | 9.26 | 12.34 | 15.43 | 18.51 | 21.60 | 24.68 | 27.77 | 30.85 |

## 4. Analysis of the Longitudinal Mechanical Properties of a Double-Layered Shield Tunnel

In this study, the deformations and internal forces of a single-layered lining tunnel and double-layered lining tunnel under various secondary lining construction times were compared. This comparison aimed to analyze the influence of secondary lining construction time on the longitudinal mechanical properties of shield tunnels. By introducing a reasonable degree, the reasonable construction time for the secondary lining was determined.

### 4.1. Analysis of the Longitudinal Deformations of a Double-Layered Shield Tunnel

Figure 13 presented the distribution curve depicting the vertical deformation of the segment of lined vault within the shield tunnel under varying secondary lining construction times. Observing the chart, it became evident that the vertical deformation pattern of the segment lining vault remained relatively consistent under different secondary lining construction times, with only variations in deformation magnitude. The vertical deformation curves of the segment lining vault all exhibited a U-shaped pattern. The maximum vertical deformation occurred within the central 30–70 ring, sharply diminishing towards the tunnel's left and right sides. The vertical deformation curve exhibited approximate symmetry, with the axis of symmetry situated at the 50th and 51st rings. Under the same load, the vertical deformation value of the single-layered segment lining vault was the largest. After the construction of the secondary lining, the deformation of the segment lining vault was restrained to varying degrees. Specifically, the earlier the construction of the secondary lining, the better the effect of restraining deformation. It was observed that the effect of constrained deformation significantly improved when the construction time was advanced from 10% to 0% compared to other construction times.

As shown in Figures 13 and 14, the vertical deformation of the tunnel vault decreased significantly immediately after the construction of the secondary lining. The vertical deformation of the 30th, 40th, and 50th rings of the segment lining was reduced by 18.22%, 16.81%, and 16.41%, respectively, under 0% construction time. At that time, the restraint effect on segment lining deformation was the best. When the construction time changed from 0% to 10%, the vertical deformation increased greatly. However, when the construction time was delayed from 10% to 20%, the vertical deformation of the segment lining did not change significantly, and a "platform" appeared in the vertical deformation curve. Subsequently, with the delay in the construction time of the secondary lining, the vertical deformation of the tunnel increased slowly at a steady rate. When the construction time of the secondary lining was 90% and 100%, the maximum vertical deformation of the double-layer-lined tunnel was 98.2% and 99.9% of that of the single-layered lining tunnel, respectively. This is close to the deformation of the single-layered lining tunnel. This observation indicates that the deformation of the segment lining had been fully released

under these construction times, and the secondary lining bore less extrusion load after construction, thereby not fully utilizing its own bearing capacity.

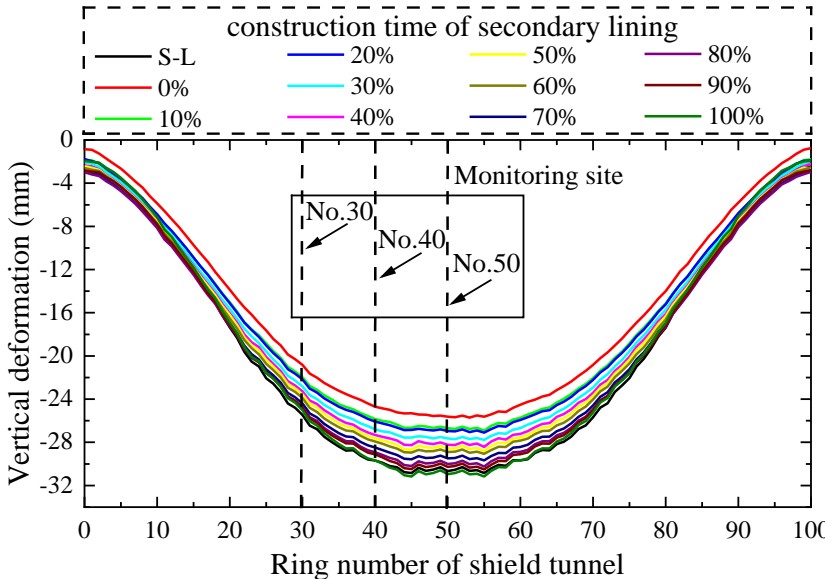

**Figure 13.** Distribution curve of vertical deformation of shield tunnel at different construction time of secondary lining.

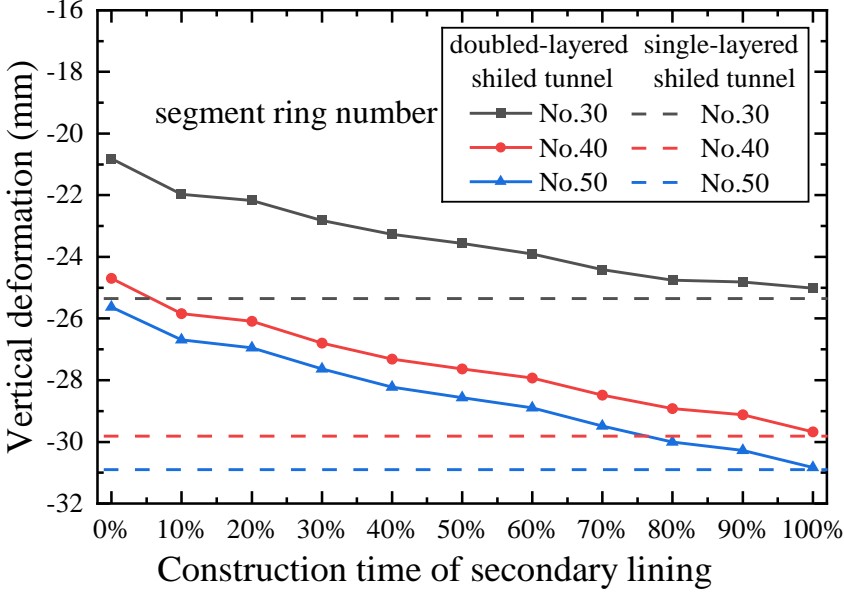

**Figure 14.** Vertical deformation curve of segment ring under different construction time of secondary lining.

Segment dislocation served as an indicator to depict the extent of non-uniform deformation within the shield tunnel. The greater the dislocation value, the more pronounced the non-uniform deformation became. Figure 15 showed the longitudinal distribution of segment dislocation in a shield tunnel under different secondary lining construction times. Given the discernible discrete nature of segment dislocation distribution, in order to effectively portray the alteration pattern of segment dislocation deformation under varying secondary lining construction times, the fitting curve for segment dislocation deformation was found, as seen in Figure 15. Each fitted curve represented a cubic function with an $R^2$ value exceeding 8.5. It was evident that the longitudinal distribution pattern of dislocation deformation remained largely consistent across different secondary lining

construction times, differing primarily in magnitude. The single-layered lining tunnel had the largest amount of dislocation, and after the construction of the secondary lining, the maximum amount of dislocation decreased significantly. Furthermore, with the advancement of the construction time of the secondary lining, the maximum dislocation value decreased gradually. The distribution curves of dislocation deformation took the form of cubic functions. In the segments spanning from ring 0 to 30 and from ring 70 to 100 on both sides of the tunnel—representing the portions of the tunnel model subject to uniform load changes—the dislocation values of the segment lining exhibited elevated levels. This indicates that when the shield tunnel experienced uneven loading, the seriousness of segment dislocation became more pronounced. Within the central region of the tunnel, owing to the even load distribution, despite a notable vertical deformation of the segment lining, the deformation across adjacent segments remained essentially identical, resulting in minimal dislocation values. Near the 50th ring of the tunnel, the dislocation value of the segment lining was virtually sustained at around 0. It is noteworthy that as the secondary lining construction time was delayed from 0% to 100%, the coefficient of determination for the fitting function of the dislocation distribution curve progressively decreased from 0.9414 to 0.8550. This signifies that as the secondary lining construction was postponed, the degree of dispersion in the dislocation distribution increased, leading to a more pronounced non-uniform deformation of the tunnel. When considered in conjunction with Figure 13, the dislocation deformation curve depicted in Figure 15 largely mirrored the first derivative of the corresponding vertical deformation curve in Figure 13. This alignment adheres closely to the definition of dislocation, consequently affirming the accuracy of the simulation outcomes presented in this paper.

As depicted in Figure 16, the segment rings with the largest dislocation deformation in regions a, b, and c were designated as monitoring points. A dislocation deformation curve was plotted to showcase the variations resulting from different secondary lining construction times. The figure illustrates that subsequent to the implementation of the secondary lining, there were varying degrees of reduction in the dislocation values of the segments. Upon considering a secondary lining construction time of 0%, the dislocation values of the segment rings with the largest dislocation deformation in regions a, b, and c exhibited reductions of 22.9%, 21.9%, and 32.2%, respectively, when compared to those of the single-layered lining tunnel. This indicated that constructing the secondary lining at that time effectively mitigated the uneven deformation of the shield tunnel. Subsequently, as the secondary lining construction time was delayed, the dislocation values of the segments increased gradually, yet they consistently remained smaller than those of the single-layered lining tunnel. When the construction time reached 80%, the dislocation values of the three segment rings reached 95.1%, 95.8%, and 97.4% of the final dislocation value observed in the single-layered lining tunnel. These values are relatively close to the ultimate dislocation deformation experienced by the single-layered lining tunnel.

In summary, the secondary lining could significantly enhance the longitudinal integrity of the shield tunnel and mitigate tunnel deformation. Moreover, the sooner the secondary lining was applied, the more pronounced its effects became. This phenomenon occurred because, when the secondary lining was timely implemented, the segment lining experienced less deformation and the pressure of the surrounding rock was not released. As a result, the secondary lining could engage in the collective deformation process at an earlier stage, assuming a greater load-bearing role on behalf of the shield tunnel. This contributed to the reduction of shield tunnel deformation. On the other hand, when the secondary lining was installed at a later stage, it was primarily regarded as a provision for structural safety and waterproofing. Its influence on the mechanical properties of the shield tunnel during excavation was relatively limited.

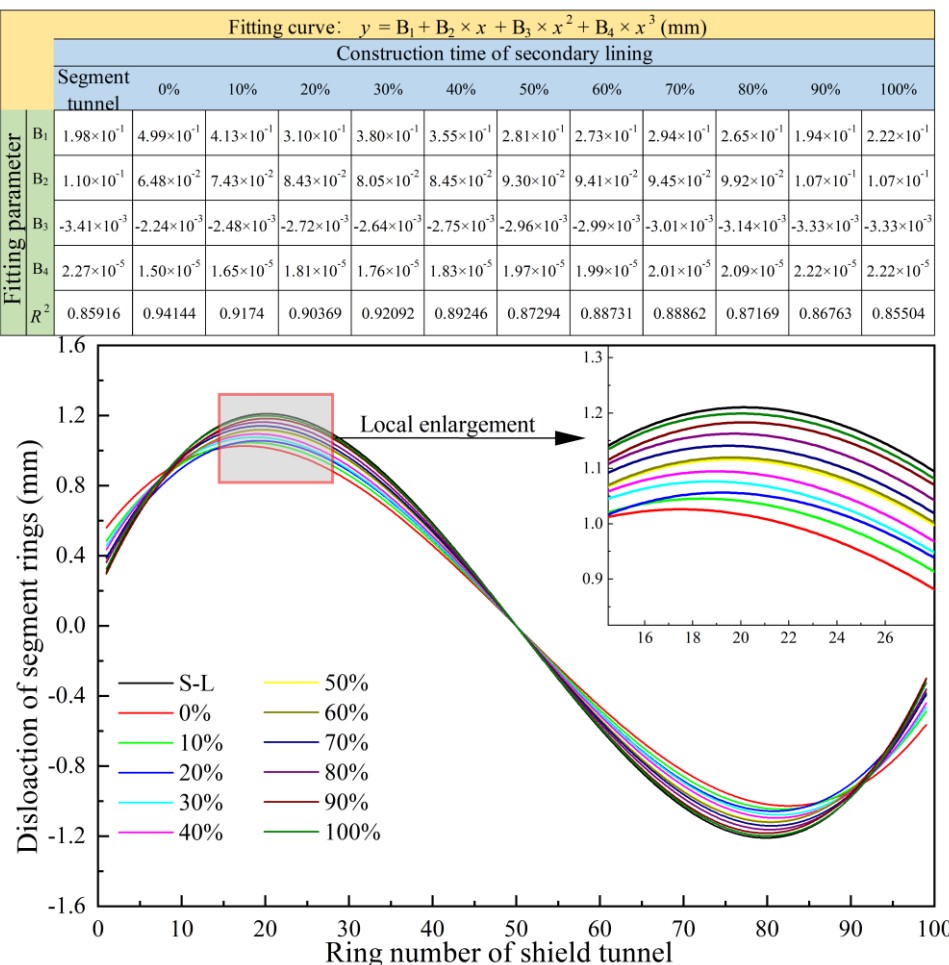

| Fitting curve: $y = B_1 + B_2 \times x + B_3 \times x^2 + B_4 \times x^3$ (mm) | | | | | | | | | | | |
|---|---|---|---|---|---|---|---|---|---|---|---|
| | | Construction time of secondary lining | | | | | | | | | | |
| Segment tunnel | | 0% | 10% | 20% | 30% | 40% | 50% | 60% | 70% | 80% | 90% | 100% |
| Fitting parameter | $B_1$ | $1.98 \times 10^{-1}$ | $4.99 \times 10^{-1}$ | $4.13 \times 10^{-1}$ | $3.10 \times 10^{-1}$ | $3.80 \times 10^{-1}$ | $3.55 \times 10^{-1}$ | $2.81 \times 10^{-1}$ | $2.73 \times 10^{-1}$ | $2.94 \times 10^{-1}$ | $2.65 \times 10^{-1}$ | $1.94 \times 10^{-1}$ | $2.22 \times 10^{-1}$ |
| | $B_2$ | $1.10 \times 10^{-1}$ | $6.48 \times 10^{-2}$ | $7.43 \times 10^{-2}$ | $8.43 \times 10^{-2}$ | $8.05 \times 10^{-2}$ | $8.45 \times 10^{-2}$ | $9.30 \times 10^{-2}$ | $9.41 \times 10^{-2}$ | $9.45 \times 10^{-2}$ | $9.92 \times 10^{-2}$ | $1.07 \times 10^{-1}$ | $1.07 \times 10^{-1}$ |
| | $B_3$ | $-3.41 \times 10^{-3}$ | $-2.24 \times 10^{-3}$ | $-2.48 \times 10^{-3}$ | $-2.72 \times 10^{-3}$ | $-2.64 \times 10^{-3}$ | $-2.75 \times 10^{-3}$ | $-2.96 \times 10^{-3}$ | $-2.99 \times 10^{-3}$ | $-3.01 \times 10^{-3}$ | $-3.14 \times 10^{-3}$ | $-3.33 \times 10^{-3}$ | $-3.33 \times 10^{-3}$ |
| | $B_4$ | $2.27 \times 10^{-5}$ | $1.50 \times 10^{-5}$ | $1.65 \times 10^{-5}$ | $1.81 \times 10^{-5}$ | $1.76 \times 10^{-5}$ | $1.83 \times 10^{-5}$ | $1.97 \times 10^{-5}$ | $1.99 \times 10^{-5}$ | $2.01 \times 10^{-5}$ | $2.09 \times 10^{-5}$ | $2.22 \times 10^{-5}$ | $2.22 \times 10^{-5}$ |
| | $R^2$ | 0.85916 | 0.94144 | 0.9174 | 0.90369 | 0.92092 | 0.89246 | 0.87294 | 0.88731 | 0.88862 | 0.87169 | 0.86763 | 0.85504 |

**Figure 15.** Longitudinal dislocation distribution curve for segment rings at different secondary lining construction times.

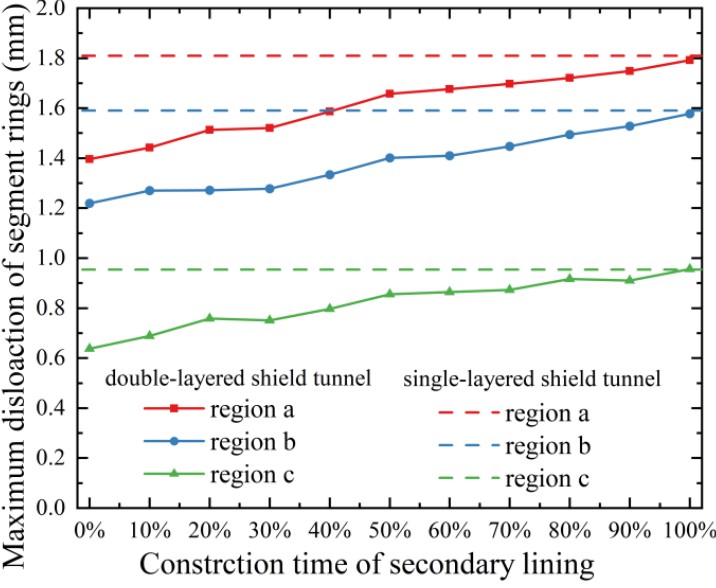

**Figure 16.** Dislocation deformation curve of segment ring under different construction time of secondary lining.

### 4.2. Analysis of Longitudinal Internal Forces

The deformation of, and internal forces on, the tunnel structure under external loads were interrelated, and they were found to be critical to the safety and stability of this shield tunnel. Therefore, apart from the deformation, it is necessary to analyze the longitudinal internal forces of double-layered shield tunnels under external load. In general, the longitudinal bending moment and shear force exert a greater influence on the tunnel structure than the longitudinal axial force [46]. Consequently, this paper primarily addressed the longitudinal bending moment and shear force of the shield tunnel, focusing on their distribution range and maximum values as the focal point of investigation. The outcomes of the calculations regarding the longitudinal internal forces within the shield tunnel under various secondary lining construction times are depicted in Figure 17.

The longitudinal internal forces in single-layered and double-layered shield tunnels have the same distribution under load, being mainly concentrated in the mid-section of the tunnel. This is consistent with the principle that the vertical deformation of the shield tunnel is large in the middle and small at both sides. Following the completion of the secondary lining construction, a noticeable alteration occurred in the internal force of the segment. This was attributed to the shared load-bearing role assumed by the secondary lining. As depicted in Table 6, the simultaneous construction of the segment and the secondary lining led to a substantial reduction in the maximum longitudinal bending moment of the segment, decreasing by 17.2% from 425.5 KN·m to 352.3 KN·m. Simultaneously, the maximum longitudinal shear force of the segment experienced a 26.8% increase from 1996 KN to 2531 KN. As the secondary lining construction time was delayed, a gradual restoration of the maximum values of the two longitudinal internal forces was observed, eventually returning to the levels observed prior to the implementation of the secondary lining's construction. In addition it could be observed that the longitudinal internal forces of the segments changed gradually during the construction time of 0% to 30%, while changing rapidly during the period of 30% to 100%. At a secondary lining construction time of 80%, the maximum longitudinal bending moment and shear force of the segments had rebounded to 96.3% and 104.0%, respectively, of the corresponding maximum values seen when the secondary lining was not constructed. This alignment brought the values closer to the internal forces experienced in a single-layered lining tunnel.

**Table 6.** Maximum value of longitudinal internal force on the segment lining.

| Type of Shield Tunnel | Construction Time of the Secondary Lining | Longitudinal Bending Moment /(kN·m) | Longitudinal Shear Force /kN |
|---|---|---|---|
| Single-layered shield tunnel | - | 425.5 | 1996 |
| Double-layered shield tunnel | 0% | 352.3 | 2531 |
| | 10% | 354.2 | 2469 |
| | 20% | 355.4 | 2412 |
| | 30% | 360.2 | 2368 |
| | 40% | 377.3 | 2293 |
| | 50% | 389.0 | 2212 |
| | 60% | 389.4 | 2160 |
| | 70% | 399.8 | 2114 |
| | 80% | 409.7 | 2075 |
| | 90% | 412.8 | 2032 |
| | 100% | 429.6 | 2018 |

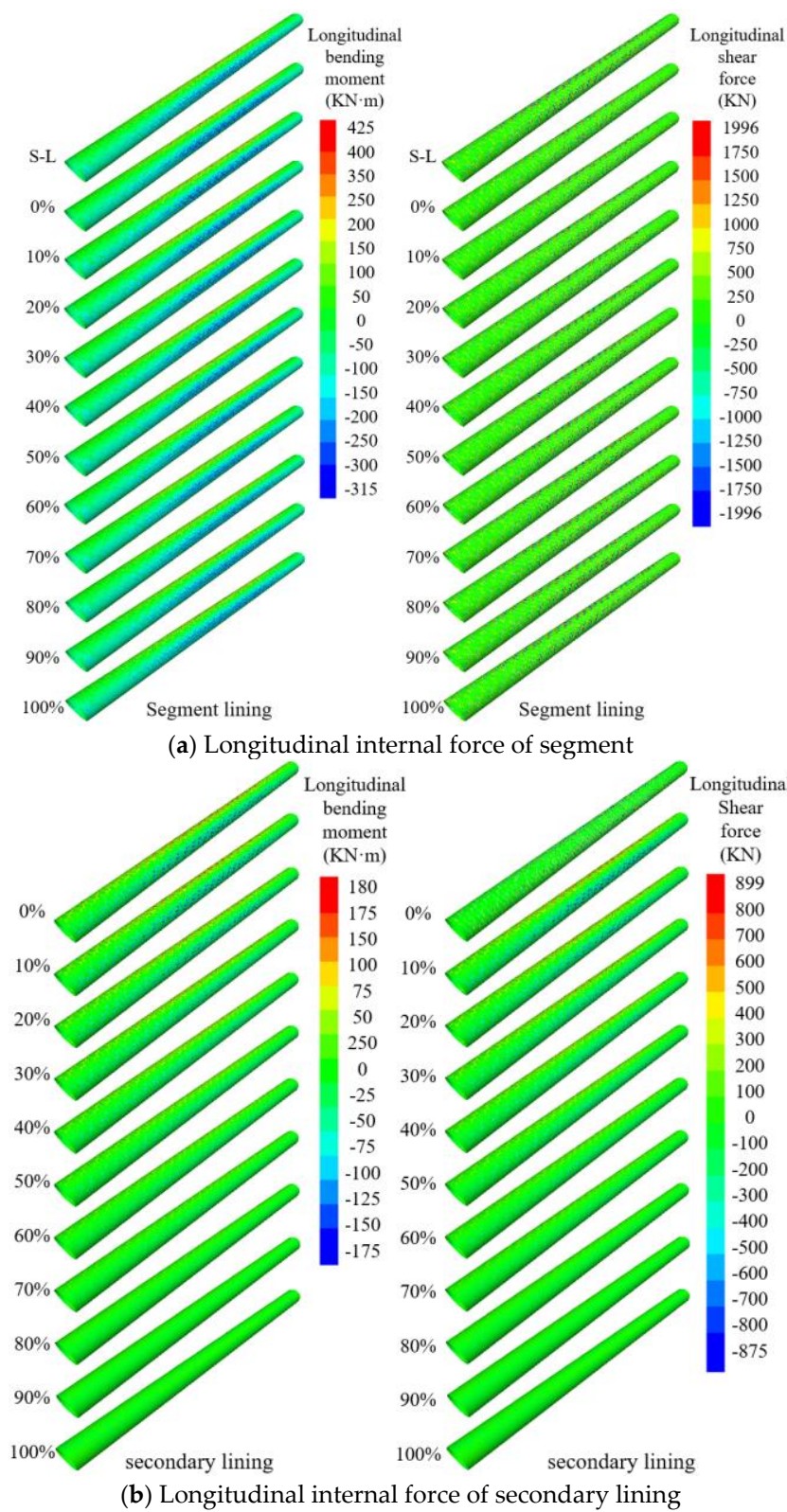

(**a**) Longitudinal internal force of segment

(**b**) Longitudinal internal force of secondary lining

**Figure 17.** Cloud diagram of longitudinal internal force of doubled-layered lining shield tunnel.

The distribution pattern of internal forces in the secondary lining resembled that of the segment, predominantly concentrated in the middle section of the tunnel. Additionally, the internal forces in the secondary lining were comparatively smaller than those in the segment. As depicted in Table 7, with a construction time of 0%, the maximum longitudinal bending moment and shear force in the secondary lining were 35.5% and 51.2% of the

corresponding values in the segment. As the construction of the secondary lining was delayed, this proportion continued to decrease. When the secondary lining construction time was 100%, the longitudinal bending moment and shear force in the secondary lining reached their minimum values, measuring 7.231 KN·m and 49.64 KN, respectively. This observation suggests that during this phase, the secondary lining's contribution to the overall tunnel force was minimal. In such a scenario, constructing the secondary lining would serve primarily as a safety reserve and waterproofing measure.

**Table 7.** Maximum value of longitudinal internal force on the secondary lining.

| Type of Shield Tunnel | Construction Time of the Secondary Lining | Longitudinal Bending Moment /(kN·m) | Longitudinal Shear Force /kN |
|---|---|---|---|
| Double-layered shield tunnel | 0% | 180.5 | 898.8 |
| | 10% | 169.6 | 883.2 |
| | 20% | 150.0 | 701.1 |
| | 30% | 140.4 | 676.4 |
| | 40% | 117.5 | 568.1 |
| | 50% | 92.9 | 474.3 |
| | 60% | 75.7 | 387.2 |
| | 70% | 57.1 | 301.0 |
| | 80% | 44.2 | 208.6 |
| | 90% | 23.5 | 105.7 |
| | 100% | 7.2 | 49.6 |

The curve depicting the variation of the maximum longitudinal internal force of the segment and the secondary lining with the construction time of the secondary lining is illustrated in Figure 18. As evident from the diagram, the sooner the secondary lining's construction commenced, the larger the load it supported, allowing for a greater utilization of the secondary lining's bearing capacity. However, following the completion of the secondary lining construction, the changing trends of longitudinal bending moment and longitudinal shear force in the segments diverged significantly. To elaborate, an earlier initiation of the secondary lining's construction yielded a more noticeable reduction in the longitudinal bending moment of the segment, accompanied by a more pronounced increase in the longitudinal shear force. The observed phenomenon is consistent with the findings of existing shield tunnel model experiments [42]. In other words, the construction of the secondary lining led to a decrease in the longitudinal bending moment while causing an increase in the longitudinal shear force of the segments. This also implies that, when designing the segment lining for the double-layered shield tunnel, greater attention should be given to the segments' shear resistance performance. Consequently, the selection of an appropriate construction time for the secondary lining, one that optimally harnessed the secondary lining's load-bearing capacity, maintained suitable longitudinal internal force values for both the segment and secondary lining, and ensured that tunnel deformation remained within allowable limits, became a subject worthy of further investigation.

*4.3. Determination of the Optimal Construction Time of the Secondary Lining*

In this paper, the reasonable construction time for the secondary lining of the shield tunnel was determined by amalgamating tunnel deformation, the internal force of both the segment lining and secondary lining, and the reasonable degree of internal force. As depicted in Figure 13, tunnel deformation reached a minimum at 0% construction time and subsequently increased with the postponement of the secondary lining's construction. Hence, the earlier the secondary lining was constructed, the more advantageous it had

in mitigating tunnel deformation. To attain an optimal deformation control effect, the reasonable construction time of the secondary lining was defined as the period when segment displacement reduced by at least 50% post-construction. Drawing from the curve illustrating the vertical deformation of the segment concerning the secondary lining's construction time in Figure 14, it was deduced that the reasonable construction time for the secondary lining, while considering the segment's deformation control effect, fell within the range of 0% to 40%.

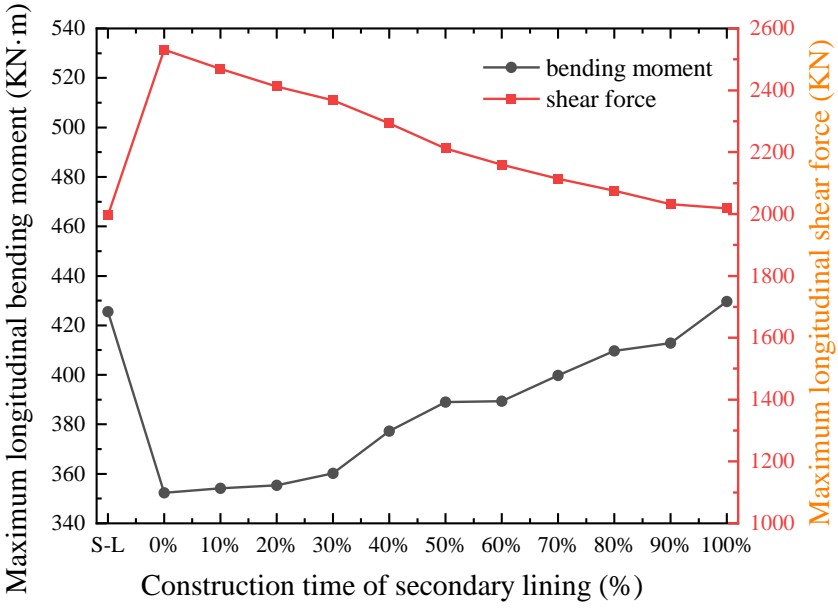

(S-L: a single-layered shield tunnel without secondary lining)

(**a**) Segment lining

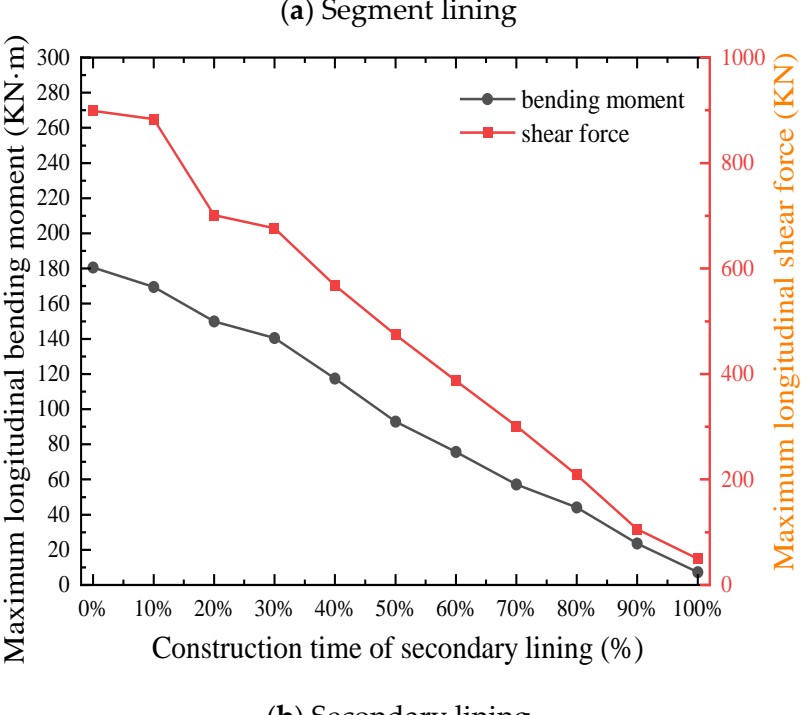

(**b**) Secondary lining

**Figure 18.** Curve of internal force of double-layered lining structure with the construction time of secondary lining.

As depicted in Figure 18a, after the secondary lining was constructed, the bending moment and shear force of the segment exhibited a distinct change pattern. Specifically, the bending moment increased while the shear force decreased. Consequently, it was essential to account for the disparity in the segment's load-bearing capacity concerning both bending moment and shear force. This study adopted the total safety factor method and introduced the reasonable degree *W* of internal force as the evaluation criterion. In contrast to the longitudinal axial force within the tunnel, the longitudinal bending moment and shear force wielded a more significant influence on the tunnel's structure, leading to more prevalent instances of damage [46]. Hence, this paper employed the longitudinal bending moment and shear force as the key indicators for calculations.

$$W = (1 - \frac{n'}{n})^{\frac{1}{3}} + (1 - \frac{m'}{m})^{\frac{1}{3}} \tag{9}$$

*n* and *m* represented the maximum longitudinal bending moment and shear force achievable by segments during various construction times, while *n'* and *m'* denoted the corresponding maximum values at a specific construction time. Subsequently, the alteration trend of the segment's internal force reasonable degree *W* with varying construction times of the secondary lining was depicted in a graph (Figure 19). A heightened reasonable degree of internal force indicated a more judicious distribution of internal forces within the segment. Figure 19 illustrated that the internal force reasonable degree peaked within the 20~100% construction time range. During this interval, the segment displayed a well-balanced internal force distribution, devoid of substantial shear forces or bending moments. This scenario showcased the segment's optimal utilization of its bending and shear resistance capacities.

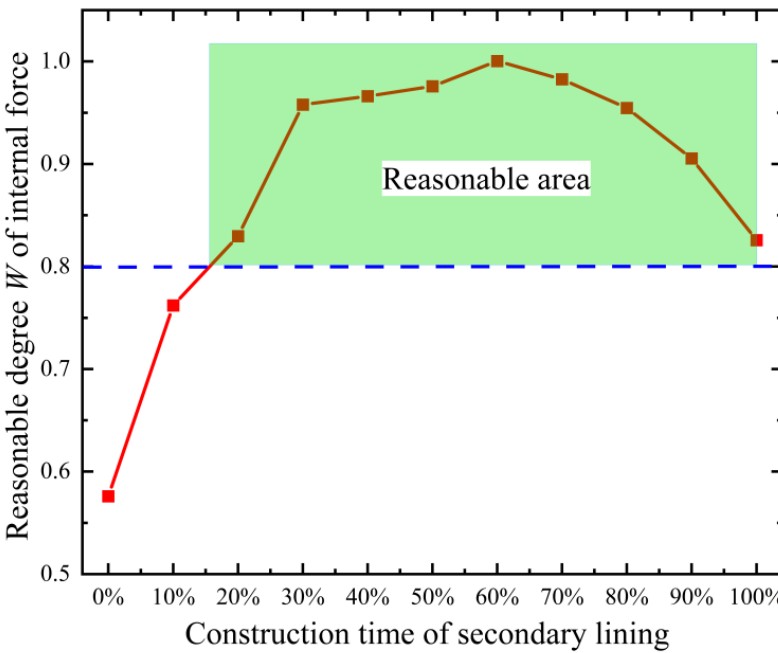

**Figure 19.** Variation in the reasonable degree of times of construction of the secondary lining.

The curve depicting the change in longitudinal internal force of the secondary lining with respect to its construction time is illustrated in Figure 18b. It can be observed from the figure that as the construction time was extended, the internal force of the secondary lining experienced a continuous reduction. This reduction indicated a decreasing contribution of the secondary lining to the overall structural forces of the tunnel. In order to fully utilize the load-bearing capability of the secondary lining, it was recommended to ensure that the internal force within the secondary lining exceeded 20% of the internal force within the

segment lining before the secondary lining was constructed. This guideline aligned with a construction time between 0% and 50%.

In conclusion, taking into account three factors, segment deformation, the internal force of the secondary lining, and the reasonable degree of segment internal force, the optimal construction time for the secondary lining was determined to be within the range of 20% to 40%. In the engineering context, this implies that commencing the construction of the secondary lining on the day when the deformation of the segment lining reaches 20% of the final deformation of a single-layered shield tunnel is reasonable, with the latest acceptable starting point being no later than 40%. Within this range, tunnel deformation remains relatively minimal, the internal force of the segment lining is reasonably distributed, and the secondary lining can effectively showcase its structural performance.

## 5. Conclusions

The effects of the secondary lining's construction time on the longitudinal mechanical properties of double-layered shield tunnels were investigated by using numerical simulation methods, and the optimal construction time of the secondary lining was determined by considering structural reliability and construction economics. The following conclusions were obtained:

(1) The construction of the secondary lining effectively reduced the vertical deformation and dislocation deformation of the segments, enhancing the integrity of the double-layered lining tunnel. However, this effect gradually diminished as the construction time of the secondary lining was delayed;

(2) The vertical deformation of the tunnel was predominantly concentrated at the position of maximum load, specifically near the middle of the tunnel. Conversely, the dislocation deformation primarily occurred in areas with uneven loading, namely along both sides of the tunnel. The alteration curve of segment dislocation essentially mirrored the first derivative of the vertical deformation curve of the segment;

(3) Following the construction of the secondary lining, the longitudinal shear force in the segments notably increased and the longitudinal bending moments significantly decreased. However, as the construction time of the secondary lining was delayed, the longitudinal internal forces within the segments gradually reverted to their levels prior to the secondary lining construction. The longitudinal internal forces within the secondary lining diminished progressively from their maximum value to zero, in tandem with the postponed construction of the secondary lining;

(4) Regarding the longitudinal deformation behavior, the early installation of the secondary lining resulted in the secondary lining assuming a higher load-bearing role. This subsequently led to diminished deformation within the segment lining. Nevertheless, this early construction also brought about an underutilization of the segment lining's load-bearing potential, ultimately resulting in the inefficient utilization of the structure's performance. Therefore, combined with the internal force regime on the double-layered lining structure, the reasonable degree of internal force of the secondary lining was introduced as the judgment criterion, and the reasonable construction time was determined to be 20% to 40% by considering the structural reliability and construction economy, meaning that the secondary lining was constructed when the deformation of the segmental lining was between 20% to 40% of the final deformation of the single-layered shield tunnel.

The research findings of this paper can be applied in the field of shield tunneling, serving as a reference for the design and construction of shield tunnels. Due to the limited availability of relevant experiments and engineering data for this topic, a significant portion of the research has relied on numerical simulations. In subsequent studies, endeavors will be made to undertake model experiments, thereby delving more deeply into this topic.

**Author Contributions:** Conceptualization, S.C. and B.H.; funding acquisition, S.C., J.C. and X.F.; investigation, Y.Y., J.H. and Y.Z.; methodology, S.C., B.H. and H.W.; resources, J.C. and X.F.; supervision, Y.Z.; validation, X.F.; writing—original draft, B.H., H.W. and J.H.; writing—review & editing, Y.Y. All authors have read and agreed to the published version of the manuscript.

**Funding:** This research was funded by the National Natural Science Foundation of China (Grant No. 52079135), the Natural Science Basic Research Plan in Shaanxi Province (No. 2020JM-234), and the Key Laboratory of Hydraulic and Waterway Engineering of the Ministry of Education and the Fundamental Research Funds for the Central Universities, CHD (No. 300102282201).

**Data Availability Statement:** The datasets used and/or analyzed during the current study are available from the corresponding author on reasonable request.

**Conflicts of Interest:** The authors declare no conflict of interest.

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
