# Peer review of "The Optimization of Secondary Lining Construction Time for Shield Tunnels Based on Longitudinal Mechanical Properties"

_applsci, doi:10.3390/app131910772_

Round 1

Reviewer 1 Report (New Reviewer)

REVIEW

on article

Optimization of Secondary Lining Construction Time for Shield Tunnel Based on Longitudinal Mechanical Properties

Shaobo Chai, Yifan Yan, Bo Hu, Hongchao Wang, Jun Hu, Jian Chen, Xiaodong Fu and Yongqiang Zhou

SUMMARY

The article submitted for review is devoted to a topical issue. The secondary lining construction time for shield tunnel was optimized based on the longitudinal mechanical properties. The relevance of the study lies in the fact that it is necessary to reduce the probability of local damage and water leaks in the counting tunnel caused by the longitudinal uneven fit of segment rings. Nowadays, the method of reducing the secondary lining is known. However, a reasonable time frame for the construction of the secondary lining is not clear.

This is an urgent engineering problem, and the solution of such a problem is quite acute in modern tunneling. In this regard, the authors conducted a number of studies on the designs of the secondary lining and obtained important results that were ultimately satisfactory for the requirements of engineering economy and structural reliability. Thus, taking into account the study, the reviewer notes a certain scientific novelty of this article and at the same time emphasizes the high practical significance of the study. Thus, according to the reviewer, the article is a high-quality, and can be recommended for publication in the journal "Applied Sciences".

At the same time, the reviewer had a number of comments. They need to be corrected. They are listed below.

COMMENTS

1. The beginning of the abstract is not well chosen. The authors should put here a clear formulation of the scientific problem. For example, to indicate the unpredictability of the timing and economic costs when creating a secondary lining, or to indicate the imperfection of existing methods for preventing or reducing the probability of local damage and water leaks in counting tunnels. This problem should be formulated at the beginning of the abstract. Now I see only engineering task, not scientific.

2. Perhaps the authors described the methodology of the study in too much detail in the middle part of the abstract. This part can be shortened somewhat so that the abstract looks like this: scientific problem, applied methodology, scientific result. In this case, the result should be displayed in a qualitative and quantitative aspect.

3. The introduction provided by the authors is not detailed enough. The authors should have worked more deeply with references devoted to tunneling, issues of tunnel lining, as well as issues of accidents and local damage and leaks of such tunnels. There are quite a lot of studies on this topic, which I recommend to mention more in the "Introduction" section.

4. The "Introduction" section should end with a clear statement of the scientific problem, scientific novelty, goal and tasks of the study.

5. Since the article looks rather large and serious in volume, it is recommended to present the experimental research program at the end of the "Introduction" section: how the authors worked, what studies were carried out, separately identify the numerical approach, the numerical model, and present all this in the form of a block diagram conducting research.

6. The authors approached very scrupulously from the point of view of the engineering task, but I would recommend to get methodological, more structured studies. Such block diagrams would be superfluous. This article would have benefited.

7. Some Figures are not of high enough quality. Figure 2 seems interesting, showing the schematic diagram of a longitudinal geological section, but I would like to see a more detailed explanation of this diagram.

8. Some graphic elements are presented in low quality. This should be corrected so that they do not have unreadable characters.

9. It seems that the authors have typos in the words, for example, in Figure 12 the letter “u” is missing in the word “Construction” and in Figure 13 there is an error in the caption “Construction Time”: it is written with an “a”, but it should be through "u". Therefore, detailed proofreading and editing is necessary in terms of style and language. Perhaps the help of an English-language editor is required. Authors need to be careful about this.

10. The section "Discussion of the obtained results" should be separated into a subparagraph, anticipating the conclusions. A detailed comparison of the obtained results with the results of other authors should be provided. Here, the scientific novelty of the study and the practical significance of the results obtained in comparison with the practical significance of the results of previous researchers should be clearly expressed.

11. Conclusions should contain a clear formulation of the scientific result, an assessment of the degree of achievement of the research goal, the formulated scientific knowledge obtained, as well as applied recommendations for the application of research. It would not be superfluous to reflect the prospects for the development of this study in the future: whether the authors will continue it or whether this article is the final one in this direction. I would also like to see a mention of this.

12. 35 references for such a topic as "Local damage to tunnels" is not enough. In addition, the reviewer draws attention to the fact that many references are older than the last five years. Tunneling around the world is developing quite intensively. You should add at least 10-12 references for the last 5 years.

In general, the reviewer believes that the article is done at a fairly high level and the study is very large scale. After correcting the remarks made, the article can be recommended for publication in the Journal "Applied Sciences".

Moderate editing of English language required

Author Response

Reviewer 2 Report (New Reviewer)

Show the delay in construction of secondary lining in days rather than percentages.

elaborate the method of modeling interfaces of primary lining and secondary lining as the Authors suggested a delay in the construction of secondary lining. There will be a cold joint/bond that needs to be addressed while analyzing through software.

what would be the effect of delamination between the main tunnel and lining during bending/uneven deformation which is not addressed in the study?

This paper is more interested in the model that could be developed for the existing double-lining tunnel to check the correctness of the model. Also, some physical tests should also be performed to confirm the finding through simulation.

Please check English and grammar.

Author Response

Reviewer 3 Report (New Reviewer)

In the present study, the authors have analyzed the influence of the construction time of the secondary lining in terms of the longitudinal mechanical properties of double-layered shield tunnels. The optimal time of construction of the secondary lining was determined using three-dimensional analysis. Please find below points that may help the authors to improve the quality of the manuscript.

1.     The captions for Figures 1, 2, and 3 may be improved. Brief details may be added.

2.     The Mohr-Coulomb constitutive model has been adopted for modeling. Suggest discussion.

3.     The picture clarity and description may be improved in Figure 4.

4.     As mentioned in the line no. 250, the link element has been represented consisting of three elastic springs. What will happen if the link or bolts yield considering the worst scenario?

5.      Figure 7-(a), (b), and (c) to be indicated properly below the Figures.

6.     Was a sensitivity analysis performed to confirm that model convergence was achieved with the selected mesh size?

7.     What boundary conditions have been applied in the present numerical study?

8.     Typo error in Figure 15.

9.     Line no. 581-583, the maximum longitudinal bending moment of the segment decreased to 350 kN from 425 kN and the maximum longitudinal shear force of the segment increased to 2530 kN from 1996 kN. Will it affect the design of the tunnel? Suggest discussion.

10.   References are to be checked thoroughly.

It can be improved.

Round 2

Reviewer 1 Report (New Reviewer)

All my comments were taken into account and sufficient corrections were done.

I recommend the article for publishing.

This manuscript is a resubmission of an earlier submission. The following is a list of the peer review reports and author responses from that submission.

Round 1

Reviewer 1 Report

Determination of the reasonable construction time of secondary lining of shield tunnel

The paper is related to a construction of secondary lining of a segment lined tunnel. To analyse the behaviour of the tunnel, the authors built a numerical model. Figure 7 shows clearly the kind of loading conditions and boundary condition applied to the numerical model of the tunnel. The loading condition applied are completely wrong, as the tunnel is fully immersed on the rock mass and the loading conditions are very far from “vertical load”. The correct loading condition is a radial pressure along the tunnel surface, like a pipe immersed in a water. Also, the boundary condition of the left and right part of the tunnel are wrong: the z and y has ben set to be zero. But the z can be variable because the whole tunnel can deform jointly with the rock mass that is deformable too, while y can be considered zero if we refer just to a limited part of the tunnel, like in this case. Results evidenced, in fact, that the deformation is mostly concentrated in the middle of the tunnel, that is the classical scheme of a fixed-end hyperstatic beam subjected to a vertical concentrated and distributed load, with maximum momentum and deformation in the middle.

I suggest to the authors to apply a more realistic load to the model and analyse the new results obtained.

Reviewer 2 Report

The paper is well written and structured. The topic is interesting and could be suitable for publication. Even so, the following items should be clarified or enhanced before acceptation:

-          - The introduction section is too large. I suggest making it shorter

-    - The section 2 should be renamed as Numerical model or numerical approach. Although the paper is long, the reviewer advises authors to show more detail about difference finite approach used to solve the numerical model. Please add equations used (in an appendix if necessary)

-         -  Please, arrange the Figure 5

-          - There are no experimental data in this research. The authors should justify this fact

-         - Regarding the numerical model used, the soil/rock properties around tunnel seem to have no influence on several internal bending and forces in lining. Therefore, what sense does Figure 2 have?. This fact should be explained by the authors

-          - In this research, the different phases of tunnel construction have not been simulated. The authors should clarify this fact, justifying their non-influence in the numerical results

-          - The figure 18 is very interesting. The question would be: how to convert the construction time of secondary lining (in terms of % deformation) of in real time?

Reviewer 3 Report

It dt doesn't make much sense to build in a secondary lining. It is unclear how a thin unloaded concrete layer under uncertain boundary conditions will remain untracked and hence retain water-ingress.

Besides, adding a secondary lining is not a remedial action but it had to be predestined to allow the cross sectional clearance. In this case it also requires much higher excavation volumes and costly and unsustainable use of concrete. 

The methodology is interesting but the topic and outcome are in my opinion irrelevant and useless. The authors should consider this.

Please also provide a lot of details from at least 3 practical applications = real projects following this secondary lining concept and its successful implementation. 

Round 2

Reviewer 1 Report

The paper has been improved by the authors and can now be published.

Detailed comments:

Figure 2: please add a note "not in scale"

Figure 3: the secondary lining is not clearly visible

Figure 10 and 17: missing measurement units of vertical displacement

Reviewer 2 Report

Although the topic is interesting and could be suitable for publication, the authors have not adequately addressed the items suggested by the reviewer. With this revision, the paper is not suitable for publication in current version. Please see the following items:

-          Numerical aspects have to be included. Mesh, calculation phases, etc.

-          The loads used are not correct. The usually loads around the tunnel do not have that distribution. Moreover, the simulation of tunnel construction is necessary with the real geotechnical properties of Fig. 2. FLAC can simulate these phases. A numerical study without realistic loads has not sense.

-          The structural model for tunnel seems more a beam than a tunnel embedded in the ground.

In my opinion, this paper should consider more realistic loads and take into account the tunnelling process. Without this, the paper is not so relevant and should not be published.

Round 3

Reviewer 2 Report

All of the limitations/suggestions marked by the reviewer have not been adequately addressed and the article has the same limitations than the previous version. My opinion is that a scientific journal indexed in the JCR in the Q2 (Engineering) should not publish this paper because the assumptions to build the numerical model are very far from the usual loads and stress around the tunnel.
